# Association between Deep Medullary Veins in the Unaffected Hemisphere and Functional Outcome in Acute Cardioembolic Stroke: An Observational Retrospective Study

**DOI:** 10.3390/brainsci12080978

**Published:** 2022-07-25

**Authors:** Chen Ye, Junfeng Liu, Chenchen Wei, Yanan Wang, Quhong Song, Ruosu Pan, Wendan Tao, Bo Wu, Ming Liu

**Affiliations:** 1Department of Neurology, West China Hospital, Sichuan University, Chengdu 610041, China; yechen11@stu.scu.edu.cn (C.Y.); junfengliu225@outlook.com (J.L.); liushuxia0777@126.com (C.W.); yanan_wang1105@163.com (Y.W.); 18804026893@163.com (Q.S.); panruosu0601@163.com (R.P.); taowendan@wchscu.cn (W.T.); dr.bowu@hotmail.com (B.W.); 2Department of Neurology, The Affiliated Hospital of Qingdao University, Qingdao 266003, China; 3Department of Geriatrics and National Clinical Research Center for Geriatrics, West China Hospital, Sichuan University, Chengdu 610041, China

**Keywords:** deep medullary veins, functional outcome, acute cardioembolic stroke, SWI

## Abstract

Objective: To explore whether deep medullary veins (DMVs) in the unaffected hemisphere were associated with functional outcome in acute cardioembolic stroke patients. Methods: Acute cardioembolic stroke patients at a single center were retrospectively included. DMVs visibility in the unaffected hemisphere was assessed using a well-established four-grade scoring method based on susceptibility-weighted imaging (SWI): grades 0–3 (grade 0 for no visible DMVs; grade 1 for the numbers of conspicuous DMVs < 5; grade 2 for numbers raging from 5 to 10; grade 3 for more than 10). Patients were further divided into mild-to-moderate (grade 0–2) and severe DMVs (grade 3) groups. Functional outcomes were evaluated using the modified Rankin scale (mRS) score at three months. Poor outcome was defined as mRS ≥ 3. Binary logistic regression analysis was used to explore the association between DMVs grade and functional outcome. Results: A total of 170 patients were finally included. Compared with the mild-to-moderate DMVs group (149 patients), the severe DMVs group (21 patients) had higher baseline National Institutes of Health Stroke Scale (NIHSS) scores (*p* = 0.002), lower levels of admission systolic blood pressure (BP) (*p* = 0.031), and elevated rates of large infarction (*p* = 0.003). At three months, the severe DMVs group had higher mRS (*p* = 0.002). Patients in the poor outcome group (82/170, 48.2%) had older age, higher baseline NIHSS score, lower admission diastolic BP, higher rates of hemorrhagic transformation and large infarction, and an increased proportion of severe DMVs (all *p* < 0.05). After adjusting for confounders, multivariable regression analysis showed that the severe DMVs grade (adjusted odds ratio [OR] = 5.830, 95% confidence interval [CI] = 1.266–26.856, *p* = 0.024) was significantly associated with three-month functional outcomes without interaction with other potential risk factors (*p* for interaction > 0.05). Conclusions: DMVs grade in the unaffected hemisphere was independently associated with three-month functional outcome in acute cardioembolic stroke patients. Patients with severe DMVs were more likely to have a poor functional outcome at three months.

## 1. Introduction

As a major contributor to death and disability, the prevalence and incidence of ischemic stroke are increasing in China [1]. Particularly, cardioembolic stroke patients have been reported to be prone to have higher frequencies of stroke-related complications (i.e., hemorrhagic transformation) and higher rates of case-fatality and disability than other ischemic stroke subtypes [2]. Although anticoagulant and reperfusion treatments, such as thrombolysis and recanalization (endovascular therapy), are recommended by current guidelines, nearly 1/3 of patients still have a poor prognosis during long-term follow-up [3]. Identifying patients with potentially poor functional outcomes at an early stage would facilitate the selection of more intensive therapeutic interventions during hospitalization and assist in guiding long-term management strategies in such groups.

In recent years, deep medullary veins (DMVs), which distribute in the periventricular white matter and serve as an important part of the deep cerebral veins system [4,5], have attracted much attention in the field of cerebrovascular disease. Through susceptibility-weighted imaging (SWI), DMVs can be visualized and assessed quantitatively or semi-quantitatively [6,7,8]. An abnormal DMVs sign, which was described as prominent or asymmetric DMVs in previous studies, is considered to indicate hypoperfusion, poor collateral circulation, and a relatively increasing level of deoxyhemoglobin of brain tissue during the acute stage of ischemic stroke [9,10,11,12,13,14,15]. As a novel neuroimaging marker, it may help to predict the extent of the ischemic penumbra [16,17,18] and evaluate the clinical status and prognosis of stroke patients [9,10,11,12,13,14,15,19,20]. Cytotoxic edema within and/or around the infarct lesion, infarction size growth, and insufficient collateral blood flow are considered as the underlying mechanisms associated with increased visibility of DMVs and thus affect clinical outcomes [9,21,22].

Accounting for 15–30% of ischemic stroke and resulting in a sudden occlusion in the intracranial artery [23,24], cardioembolic stroke patients may experience more dramatic changes in microcirculation status and have poorer establishments of collateral circulations [25,26]. Both the pathophysiological processes can affect the appearance of DMVs, as mentioned above. Additionally, a recent prospective study indicated that cardio-embolism might be independently associated with the prominent vessel sign on SWI [27]. It was proposed that cardioembolic thrombus of medium to large size could both lead to the presence of prominent DMVs under the circumstance of acute cerebral parenchymal and/or cortical ischemia. Generally, patients with cardioembolic stroke are more likely to suffer large infarcts, greater stroke severity (higher admission NIHSS (National Institutes of Health Stroke Scale)), and worse functional outcomes than patients with non-cardioembolic stroke [1,3,24,28]. To date, little is known about the association between DMVs and clinical outcomes of acute cardioembolic stroke patients. Considering the potential influence of infarction lesion and edema on DMVs, the aim of this study was to explore the relationships between DMVs in the unaffected hemisphere and functional outcome in acute cardioembolic stroke patients.

## 2. Methods

### 2.1. Study Participants

Consecutive acute cardioembolic stroke patients (time from onset to admission ≤ 7 days) admitted to West China Hospital, Sichuan University from October 2013 to June 2018 were retrospectively retrieved in this study. Patients were included in the study if they: (1) had a clinical diagnosis of first-ever acute ischemic stroke according to the World Health Organization criteria [29] and unilateral anterior circulation infarction confirmed by following magnetic resonance imaging (MRI) examinations, (2) had completed essential examinations such as electrocardiogram, echocardiography, cervical vascular ultrasound, and cranial computerized tomography or MR angiography (CTA/MRA) and been identified as cardioembolic stroke subtype defined by the Trial of Org 10,172 in Acute Stroke Treatment (TOAST-criteria) [30], and (3) had completed MRI sequences including SWI. Patients were excluded for the following reasons: (1) quality of radiological findings was poor, (2) had an intracranial hemorrhage on baseline CT scans, (3) there were bilateral infarction lesions, (4) had other abnormal brain MRI findings, such as previous stroke, tumor, trauma, venous embolism, or infection, (5) had radiographic evidence of severe stenosis other than the responsible vessel and moderate-to-severe stenosis or occlusion in the contralateral intracranial artery, (6) received reperfusion therapy, and (7) were lost to follow-up at 3 months.

### 2.2. Clinical Assessment and Follow-Up Outcome Measures

Demographic and clinical information was collected with a standardized format, including age, gender, prior medical history (hypertension, diabetes mellitus, and hyperlipidemia), previous use of antithrombotic agents (antiplatelets and anticoagulants), cardiological conditions (atrial fibrillation, rheumatic heart disease, congestive heart failure, mitral valve stenosis, and infective endocarditis), and current smoking and drinking. Baseline stroke severity was assessed by the National Institutes of Health Stroke Scale (NIHSS) on admission, blood pressure on admission, time from onset to admission, and treatments during hospitalization, including antiplatelets and anticoagulants.

The modified Rankin scale (mRS) score at 3 months was used to evaluate functional outcome in the outpatient clinic or through a structured telephone interview. A poor functional outcome was defined as mRS ≥ 3 and a good functional outcome as mRS < 3.

### 2.3. Neuroimaging Acquisition: CT and MRI

All patients completed brain CT scans (including CT angiography, CTA) within 24 h after admission. Scheduled brain MRI scans were obtained within 7 days after admission. Repeat CT scans were performed if hemorrhage was suspected, such as in the case of headache or other significant neurological deterioration. All MRI scans were obtained on a 3.0T scanner (Trio Siemens, Berlin, Germany). The scanning sequences included SWI, fluid-attenuated inversion recovery (FLAIR) imaging, T1-weighted imaging, T2-weighted imaging, and diffusion-weighted imaging (DWI). Detailed MRI parameters have been described in our previous studies [31,32].

### 2.4. Assessment of DMVs Grade and Other Neuroimaging Characteristics

A four-grade scoring method ranging from 0 to 3 based on minimum-intensity projection imaging (MIP) of the SWI was used to assess the DMVs grade in the unaffected hemisphere, which were shown as linear hypointense vessels [7]. The numbers of the most conspicuous and continuous medullary venous were used to evaluate grades of DMVs: 0 for no visible conspicuous and continuous veins, grade 1 for the numbers of conspicuous DMVs < 5, grade 2 for numbers raging from 5 to 10, and grade 3 for numbers more than 10 (shown in Figure 1). Then, the DMVs grade was further classified as mild-to-moderate (grade 0–2) and severe (grade 3). Considering the need for covering most of the drainage area of DMVs [5], consecutive periventricular imaging slices (nearly 10 mm-thick, total of 5 slices) from the level of the ventricles immediately above the basal ganglia to the level of the ventricles immediately disappeared were manually assessed separately, based on visual inspection.

Hemorrhagic transformation (HT) was defined as hemorrhage within the infarct territory or parenchyma hemorrhage outside the infarct lesion that was detected on follow-up CT or MRI during hospitalization, but not on head CT on admission [33]. As in our previous studies [31,32,34], we further divided HT into hemorrhagic infarction (HI), parenchymal hemorrhage (PH) based on the criteria of radiographic appearance [35], and symptomatic HT (sHT) on neurological symptoms [36]. Large hemispheric infarction was recorded when the infarct area was larger than 1/2 of the blood supply area of the middle cerebral artery (MCA) on FLAIR and/or DWI [37,38].

All images were reviewed and analyzed by two trained readers (C.Y. and J.L.) blinded to the clinical information. A disagreement was solved through discussion or advice from a third experienced researcher (B.W.). Those with obvious head movements, artifacts, midline shift, intracranial mass effect, or other conditions that will result in poor image quality and affect the interpretation of the DMVs indicator were excluded in the final analysis. The inter-rater kappa coefficient was 0.86 for mild-to-moderate DMVs grade, 0.88 for severe DMVs grade, 0.86 for HT, and 0.83 for infarct lesion > 1/2 middle cerebral artery (MCA) territory.

### 2.5. Statistical Analysis

Continuous data were reported as mean ± standard deviation or median with interquartile range, while categorical data were reported as frequencies and percentages. Differences between different groups were assessed using the Student’s *t*-test or Kruskal–Wallis *H* test for continuous variables, while the chi-squared (χ2) test or Fisher’s exact test for categorical variables. Inter-rater reliability of the neuroimaging variable was tested using the kappa statistic. Binary logistic regression was applied to evaluate the independent relationship between DMVs grade and 3-month functional outcome. Variables with *p* < 0.1 in univariable analyses were included in the multivariable analysis. Stratified analyses were performed to investigate the potential effect modification by key confounding factors (age, initial stroke severity (baseline NIHSS score), anticoagulant therapy, hemorrhagic transformation, and large infarction area). The interactions between DMVs grade and confounding factors were tested via adding the multiplicative term to the multivariable logistic regression model. When appropriate, the results were reported as odds ratio (OR) with 95% confidence interval (CI).

All statistical analyses were performed using the Statistical Package for the Social Sciences 23.0 (SPSS; IBM, Armonk, NY, USA) and a two-sided *p* < 0.05 was considered statistically significant.

### 2.6. Ethical Standards and Participants’ Consent

This study was conducted in accordance with the world medical association declaration of Helsinki and was approved by the Medical Ethics Committee of West China Hospital (No. 2014(69)). Informed consent was obtained from each patient or their legal guardians.

## 3. Results

### 3.1. Baseline Information and Comparison between the Two DMVs Groups

A total of 170 acute cardioembolic stroke patients without reperfusion therapy were finally included in the present study (Figure 2). Baseline information was presented in Table 1. Among the 170 patients, the median age was 72 years old (IQR 60–78) and 40% of them were males. The baseline median NIHSS score was 8 (IQR 4–13). Of the included participants, 140 (82.4%) patients had atrial fibrillation, 42 (24.7%) patients had rheumatic heart disease, 14 (8.2%) had congestive heart failure, 14 (8.2%) had mitral valve stenosis, and 1 (0.6%) patient had infective endocarditis. After admission, 137 (80.6%) patients received treatments of an antiplatelet and 53 (31.2%) received an anticoagulant. At three months, the severe DMVs group had higher mRS scores (4 (3–4) vs. 2 (1–3), *p* = 0.002). Additionally, the distribution of mRS scores was significantly different between patients with severe DMVs and those with mild DMVs (*p* = 0.002, Figure 3).

One hundred and forty-nine (87.6%) patients had mild-to-moderate DMVs and twenty-one (12.4%) had severe DMVs (Table 1). Compared with patients with mild-to-moderate DMVs, the severe DMVs group had higher NIHSS scores (15 (6.5–18) vs. 8 (4–12), *p* = 0.002), lower SBP levels (124.6 ± 21.4 vs. 136.0 ± 22.6, *p* = 0.031) on admission, and a higher proportion with large hemispheric infarction (61.9% vs. 29.5%, *p* = 0.003). There were no significant differences found in age, gender, atrial fibrillation, rheumatic heart disease, congestive heart failure, mitral valve stenosis, infective endocarditis, and hemorrhagic transformation between the two groups.

### 3.2. Comparison of Clinical Features and Neuroimaging Characteristics between the Two Functional Outcome Groups

Eighty-two (48.2%) patients had a poor functional outcome at three months and eighty-eight (51.8%) had a good outcome (Table 2). Patients with poor three-month functional outcomes were older (75 (61–80) vs. 69 (58–77), *p* = 0.02), had higher baseline NIHSS scores (12 (8–16) vs. 5 (2–9), *p* = 0.001), lower admission DBP levels (79.1 ± 16.6 vs. 83.3 ± 14.1, *p* = 0.046), lower rates of anticoagulation treatment in hospital (28.3% vs. 71.7%, *p* < 0.001), and an increased proportion of HT (64.9% vs. 35.1, *p* = 0.001) and large hemispheric infarction (73.7% vs. 26.3%, *p* = 0.001). Additionally, patients with poor functional outcomes were more likely to have severe DMVs (22.0% vs. 3.4%, *p* < 0.001).

### 3.3. Association between Severe DMVs Grades and Three-Month Poor Functional Outcomes

As shown in Table 3, in univariable regression analysis, severe DMVs grade (OR 7.769, 95% CI 2.25–28.22, *p* = 0.001), baseline NIHSS score (OR 1.237, 95% CI 0.998–1.048, *p* < 0.001), anticoagulation in hospital (OR 0.295, 95% CI 0.146–0.594, *p* = 0.001), and presence of HT (OR 3.367, 95% CI 1.785–6.350, *p* < 0.001) and large infarction (OR 5.110, 95% CI 2.526–10.336, *p* < 0.001) were significantly associated with three-month poor functional outcomes. After fully adjusting for potential confounders with *p*-value < 0.1 in univariable analysis (i.e., age, baseline NIHSS score, admission DBP, congestive heart failure, anticoagulant treatment in hospital, and presence of HT and large hemispheric infarction), severe DMVs (OR 5.830, 95% CI 1.266–26.856, *p* = 0.024) and baseline NIHSS (OR 1.191, 95% CI 1.099–1.290, *p* < 0.001) were independently associated with three-month poor functional outcomes.

### 3.4. Stratified Analyses to Identify Variables That May Modify the Association between Severe DMVs and Three-Month Functional Outcomes

The results of stratified analyses are shown in Table 4. After adjustment for potential covariates, the association between severe DMVs and three-month functional outcome was consistent across the subgroups by age, initial stroke severity (baseline NIHSS score), anticoagulant therapy in hospital, hemorrhagic transformation, and large hemispheric infarction (all *p* for interaction > 0.05).

## 4. Discussion

The DMVs are widely investigated in patients with cerebrovascular disease, but little attention has been given to acute cardioembolic ischemic stroke patients specifically. In the present study, we explored the DMVs grade of the unaffected hemisphere in acute cardioembolic stroke patients and found that a severe DMVs score was independently associated with three-month poor functional outcome.

Since Morita et al. [6] firstly described this distinctive vessel sign in acute stroke patients, defined as the hypo-intensity and dilation of veins in the deep white matter on SWI compared to the unaffected hemisphere, the appearance of prominent or asymmetric medullary veins on the affected side was correlated with stroke severity [12,13,16,20,39,40], infarction volume [9,12,13], and poor outcome [9,10,13] in some of the previous investigations. The underlying mechanism is mainly focused on slow drainage and enlargement of veins due to increased oxygen extraction fractions (OEF) in the hypoperfusion tissue and/or cytotoxic edema within the infarct area, leading to an increased level of deoxyhemoglobin [6,7,8,22]. Additionally, poor compensation of the collateral circulation is also considered to be related to prominent DMVs and poor outcome [10,11,14,20], especially in patients with severe stenosis or occlusion of large arteries, such as the middle cerebral artery. On the contrary, better outcomes were also found in patients with asymmetry or prominent veins in some other studies, in which this imaging marker was thought to represent good collateralization and indicate an ischemic penumbra [17,19]. Different classifications and interpretations of DMVs grades, heterogeneity between patients, and various timing of MRI examinations may have led to the above differences. In this study, we only included acute ischemic cardioembolic stroke patients who were not able to receive reperfusion therapy and had no evidence of severe stenosis or occlusion other than responsible vessels, and we found a significant relationship of severe DMVs grade with three-month poor functional outcome. Our results indicate a potential role of severe DMVs as an imaging marker for clinical practice in such patients. Further studies with larger sample sizes, more subgroups with various subjects (such as groups with reperfusion treatment), and longer follow-up are needed.

Notably, in Yu et al.’s [13] study, prominent contralateral DMVs, which were assessed by comparing numbers of veins on each hemisphere, were more common in patients with good outcomes (37.5% vs. 5.3%), although it was not an independent predictor of clinical prognosis. They hypothesized that it might reflect hyper-perfusion and decreased deoxyhemoglobin concentration in the affected hemisphere due to local metabolic changes [41], thus resulting in an increase in signal intensity within the ipsilateral veins and a relatively prominent contralateral venous sign. Nevertheless, conclusions of such a comparison may be inaccurate for the influence of local edema, metabolism, or inflammation on the veins itself [5,39], especially in the case of large cerebral infarction [12]. To avoid the influence of the infarction lesion and the surrounding edema zone on the veins, in our study we directly investigated degrees of the unaffected hemispheric DMVs without a comparison to the affected hemisphere. The present results showed that the severe DMVs grade was independently associated with three-month poor outcome, which is consistent with most of the previous studies. Hemodynamic blood flow balance and compensation through the circle of Willis (CoW) between each hemisphere has been proposed and well-established [42,43]. We hypothesize that the severe contralateral DMVs may be related to a relatively more vigorous collateral flow and compensatory elevating OEF for maintaining the balance of oxygen metabolism [19] on the contralateral hemisphere due to catastrophic occlusion of a sudden embolism of cardiac origin in responsible arteries. More studies are needed to investigate the impact of structural integrity and variation of CoW, perfusion, and oxygen metabolism in each hemisphere in the association of DMVs and clinical outcomes.

Meanwhile, we also found that patients with severe DMVs had higher baseline NIHSS scores, and greater proportions of large infarction. Similarly, several previous studies have found that prominent DMVs groups had higher admission NIHSS scores and were prone to develop HT and large infarction size [13,16,27]. Resulting from the sudden decrease of cerebral blood flow, poor establishment of collateral circulation, and severe blood–brain barrier dysfunction, severe DMVs may share common pathophysiological mechanisms with hemorrhagic transformation and large infarction, along with greater stroke severity and consequently, worse clinical outcomes [44]. Besides, to mitigate the potential impact of these risk factors on clinical outcome, subgroup analyses were further performed in this study. Stratified results showed that potential confounders, including NIHSS, HT, and large infarcts, did not affect the independent relationship between severe DMVs and poor outcome, with all *p* for interaction > 0.05. Once again, it is suggested that DMVs in the unaffected hemisphere could be used as imaging markers for reflecting the damage of acute ischemia resulted from sudden arterial cardio-embolism, especially with no effect of infarction lesion, edema, inflammation, etc., on venous visualization. Moreover, our results also support the significantly independent association of DMVs with three-month functional outcome, which highlights the need to take this novel marker into consideration in the clinical management of patients with acute cardioembolic stroke.

Furthermore, most of the earlier studies only classified patients from the perspective of the responsible arteries involved (stenosis or occlusion) and did not report the specific etiological classification of stroke patients. Ischemic stroke patients with different etiology types may have varying pathophysiological changes [45], and thus the treatments are also slightly different (such as antiplatelet, anticoagulation, etc.). In this study, we found that severe contralateral DMVs are an independent risk factor for poor prognosis among cardioembolic stroke patients, who are generally more likely to suffer from more severe illnesses and higher mortality and disability rates [2]. As one of standard treatments for patients with cardioembolic stroke [46], there was a higher likelihood to have a better clinical outcome for patients receiving anticoagulation treatment after admission in the present study (unadjusted OR 0.295, 95% CI 0.146–0.594, *p* = 0.001), although there was no statistical significance in the multivariable analysis (adjusted OR 0.393, 95% CI 0.154–1.006, *p* = 0.051). Additionally, the usage of anticoagulants in China is still low, due to the consideration of complications such as hemorrhage [1,28]. Our findings not only suggest that DMVs can be considered in the management of acute cardioembolic stroke patients, but also propose the use of more intensive therapy (such as thrombolysis, anticoagulants, etc.) in such patients after weighing the benefits and risks. More clinical studies on venous dysfunction and anticoagulation in patients with acute ischemic stroke are requested in the future.

However, our study has several limitations. First, the sample size was relatively small, and our patients were included from a single hospital, which may limit the generalizability of our findings. Further studies with larger samples are required to validate these findings. Second, the status of blood flow or perfusion was not assessed due to the unavailability of relevant information. Finally, dynamic observations of DMVs were not performed in the present study, and they are worth performing in future studies.

In conclusion, DMVs in the unaffected hemisphere were significantly and independently associated with three-month functional outcome in acute cardioembolic stroke patients. Patients with severe DMVs were prone to have poor functional outcomes at three months.

## Figures and Tables

**Figure 1 brainsci-12-00978-f001:**
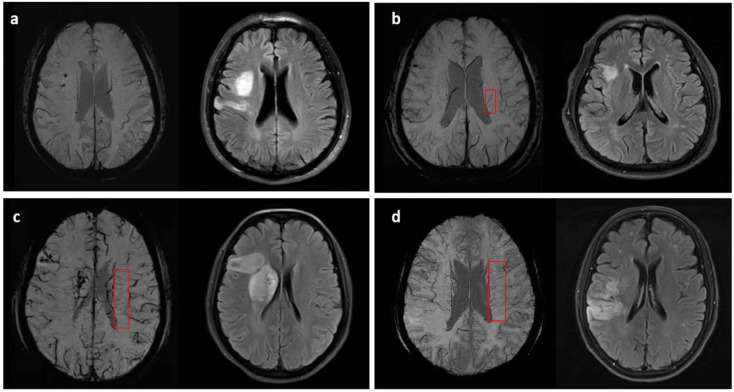
Examples and demonstrations of the four-grade DMVs (marked in the red box) scoring method on SWI with corresponding infarction lesions: (**a**) grade 0, no visible conspicuous and continuous DMVs; (**b**) grade 1, <5 conspicuous DMVs; (**c**) grade 2, 5–10 conspicuous DMVs; (**d**) grade 3, >10 conspicuous DMVs.

**Figure 2 brainsci-12-00978-f002:**
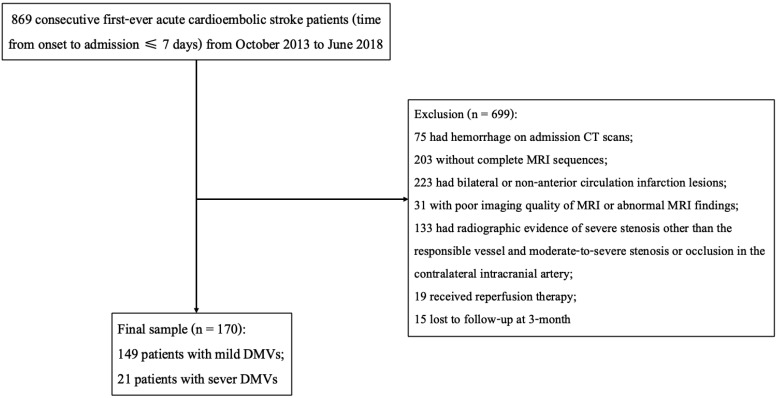
Flow chart of patient enrollment.

**Figure 3 brainsci-12-00978-f003:**
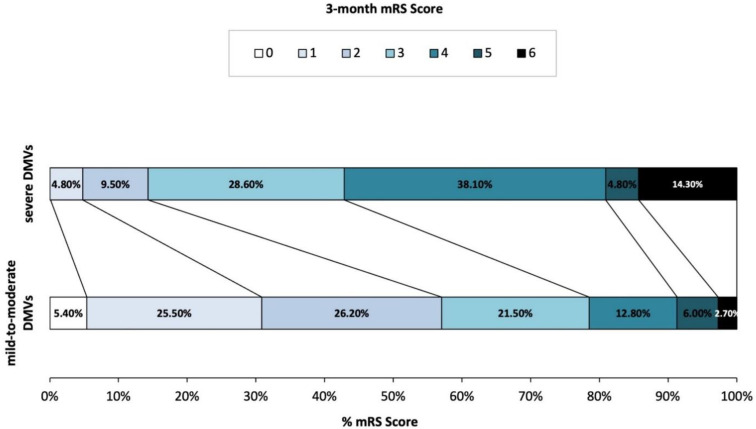
Distribution of modified Rankin scale at 3-month in patients with different DMV grades.

**Table 1 brainsci-12-00978-t001:** Baseline information and differences of variables between the two DMVs grading groups.

	All	Mild-to-Moderate DMVs	Severe DMVs	*p*
	(n = 170)	(Grades ≤ 2, n = 149)	(Grade = 3, n = 21)	
**Clinical features**				
Age (years), median (IQR)	72 (60–78)	71 (60–78)	76 (57–79)	0.519
Male, n (%)	68 (40.0)	61 (40.9)	7 (33.3)	0.505
Hypertension, n (%)	74 (43.5)	67 (45.0)	7 (33.3)	0.314
Diabetes mellitus, n (%)	49 (28.8)	45 (30.2)	4 (19.0)	0.291
Hyperlipidemia, n (%)	35 (20.6)	32 (21.5)	3 (14.3)	0.573
Atrial fibrillation, n (%)	140 (82.4)	124 (83.2)	16 (76.2)	0.539
Rheumatic heart disease, n (%)	42 (24.7)	38 (25.2)	4 (19.0)	0.601
Congestive heart failure, n (%)	14 (8.2)	10 (6.7)	4 (19.0)	0.076
Mitral valve stenosis, n (%)	14 (8.2)	14 (9.4)	0	0.221
Infective endocarditis, n (%)	1 (0.6)	1 (0.7)	0	1
Prior antiplatelet, n (%)	49 (28.8)	44 (29.5)	5 (23.8)	0.588
Prior anticoagulation, n (%)	30 (17.6)	28 (18.8)	2 (9.5)	0.376
Smoking, n (%)	35 (20.6)	31 (20.8)	4 (19.0)	1
Drinking, n (%)	26 (15.3)	22 (14.8)	4 (19.0)	0.534
Baseline NIHSS, median (IQR)	8 (4–13)	8 (4–12)	15 (6.5–18)	**0.002**
Admission SBP, mmHg, mean (SD)	134.5 (22.7)	136.0 (22.6)	124.6(21.4)	**0.031**
Admission DBP, mmHg, mean (SD)	81.3 (14.5)	81.7 (15.3)	78.4 (16.5)	0.355
Onset-to-admission time, hours, median (IQR)	24 (4–48)	24 (4–48)	5 (2.5–66)	0.262
Antiplatelet in hospital, n (%)	137 (80.6)	119 (79.9)	18 (85.7)	0.769
Anticoagulation in hospital, n (%)	53 (31.2)	46 (30.9)	7 (33.3)	0.820
3-month mRS, median (IQR)	2 (1–4)	2 (1–3)	4 (3–4)	**0.002**
**Neuroimaging characteristics**				
HT, n (%)	74 (43.5)	65 (43.6)	9 (42.9)	0.947
HI, n (%)	52 (30.6)	46 (30.9)	6 (28.6)	0.830
PH, n (%)	22 (12.9)	19 (12.8)	3 (14.3)	1
Symptomatic HT, n (%)	17 (10.0)	12 (8.1)	5 (23.8)	0.41
Large hemispheric infarction, n (%)	57 (33.5)	44 (29.5)	13 (61.9)	**0.003**

DMVs, deep medullary veins; SBP, systolic blood pressure; DBP, diastolic blood pressure; HT, hemorrhagic transformation; HI, hemorrhagic infarction; PH, parenchymal hemorrhage; SD, standard deviation; IQR, interquartile range; mRS, modified Rankin scale; NIHSS, National Institutes of Health Stroke Scale.

**Table 2 brainsci-12-00978-t002:** Differences of clinical features and neuroimaging characteristics between the two functional outcome groups.

	Good Functional Outcome (3-Month mRS < 3, n = 88)	Poor Functional Outcome (3-Month mRS ≥ 3, n = 82)	*p*
**Clinical features**			
Age (years), median (IQR)	69 (58–77)	75 (61–80)	**0.020**
Male, n (%)	39 (57.4)	29 (42.6)	0.234
Hypertension, n (%)	36 (48.6)	38 (51.4)	0.475
Diabetes mellitus, n (%)	21 (42.9)	28 (57.1)	0.139
Hyperlipidemia, n (%)	15 (42.9)	20 (57.1)	0.237
Atrial fibrillation, n (%)	75 (85.2)	65 (79.3)	0.308
Rheumatic heart disease, n (%)	21 (23.9)	21 (25.6)	0.792
Congestive heart failure, n (%)	4 (28.6)	10 (71.4)	0.070
Mitral valve stenosis, n (%)	6 (6.8)	8 (9.8)	0.486
Infective endocarditis, n (%)	0	1 (1.2)	0.482
Prior antiplatelet, n (%)	26 (53.1)	23 (46.9)	0.830
Prior anticoagulation, n (%)	17 (56.7)	13 (43.3)	0.554
Smoking, n (%)	18 (51.4)	17 (48.6)	0.964
Drinking, n (%)	16 (61.5)	10 (38.5)	0.279
Baseline NIHSS, median (IQR)	5 (2–9)	12 (8–16)	**0.001**
Admission SBP, mmHg, mean (SD)	135.3 (21.4)	133.7 (24.1)	0.646
Admission DBP, mmHg, mean (SD)	83.3 (14.1)	79.1 (16.6)	**0.046**
Onset-to-admission time, hours, median (IQR)	23.5 (4–48)	24 (3–48)	0.998
Antiplatelet in hospital, n (%)	71 (51.8)	66 (48.2)	0.975
Anticoagulation in hospital, n (%)	38 (71.7)	15 (28.3)	**<0.001**
**Neuroimaging characteristics**			
HT, n (%)	26 (35.1)	48 (64.9)	**0.001**
Large hemispheric infarction, n (%)	15 (26.3)	42 (73.7)	**0.001**
DMVs grade, n (%)			**<0.001**
Mild-to-moderate DMVs	85 (96.6)	64 (78.0)	
Severe DMVs	3 (3.4)	18 (22.0)	

DMVs, deep medullary veins; SBP, systolic blood pressure; DBP, diastolic blood pressure; HT, hemorrhagic transformation; SD, standard deviation; IQR, interquartile range; NIHSS, National Institutes of Health Stroke Scale.

**Table 3 brainsci-12-00978-t003:** Binary logistic regression analysis for associations between severe DMVs with poor functional outcomes at three months.

	Unadjusted			Adjusted		
Variable	OR	95% CI	*p*	OR	95% CI	*p*
Severe DMVs	7.769	2.25–28.22	**0.001**	5.830	1.266–26.856	**0.024**
Age ^†^	1.023	0.998–1.048	0.068	1.026	0.993–1.059	0.124
Baseline NIHSS ^†^	1.237	1.151–1.328	**<0.001**	1.191	1.099–1.290	**<0.001**
DBP on admission ^†^	0.982	0.963–1.002	0.078	0.982	0.957–1.008	0.174
Congestive heart failure	2.917	0.877–9.698	0.081	2.671	0.506–14.088	0.247
Anticoagulation in hospital	0.295	0.146–0.594	**0.001**	0.393	0.154–1.006	0.051
Presence of HT	3.367	1.785–6.350	**<0.001**	1.355	0.569–3.226	0.493
Large hemispheric infarction	5.110	2.526–10.336	**<0.001**	1.747	0.689–4.426	0.240

DMVs, deep medullary veins; DBP, diastolic blood pressure; NIHSS, National Institutes of Health Stroke Scale; HT, hemorrhagic transformation; OR, odds ratio; CI, confidence interval. ^†^ Continuous variables.

**Table 4 brainsci-12-00978-t004:** Stratified analyses to identify variables that may modify the association between severe DMVs and three-month functional outcomes.

Variable	OR	95% CI	*p*	*p* for Interaction
Age				0.610
<65	4.820	0.590–39.369	0.142	
≥65	4.371	0.442–43.209	0.207	
Baseline NIHSS				0.140
<15	8.903	1.543–51.383	0.015	
≥15	2.271	0.158–32.637	0.546	
Anticoagulation in hospital				0.998
Yes	0.850	0.092–7.859	0.886	
No	NA	NA	NA	
Presence of HT				0.332
Yes	0.920	0.072–11.709	0.949	
No	15.071	1.823–124.583	0.012	
Large hemispheric infarction				0.646
Yes	21.515	0.526–880.754	0.105	
No	3.898	0.611–24.885	0.150	

NIHSS, National Institutes of Health Stroke Scale; HT, hemorrhagic transformation; OR, odds ratio; CI, confidence interval. Adjusted for the same variables as multivariable analysis in Table 3, except for the stratified variable.

## Data Availability

The data that support the findings of this study are available from the corresponding author upon reasonable request.

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
