# Peer review of "Association between Deep Medullary Veins in the Unaffected Hemisphere and Functional Outcome in Acute Cardioembolic Stroke: An Observational Retrospective Study"

_brainsci, 2022, doi:10.3390/brainsci12080978_

Round 1

Reviewer 1 Report

This retrospective single-center study aimed to explore the relationships between deep medullary veins (DMVs) in unaffected hemisphere and functional outcome in acute cardioembolic stroke patients. They found that DMVs grade was independently associated with 3-month functional outcome in acute cardioembolic stroke patients and also showed that patients with severe DMVs were more likely to have poor functional outcome at 3-month. The authors wrote the study results and conclusions well using appropriate statistical methods. Here are my suggestions:

1.       In Introduction, the rationale for why cardioembolic stroke patients were selected for the study is insufficient.

2.       Excluding patients who meet the exclusion criteria may bias the results of the analysis. How many patients meet each exclusion criterion? To clarify this, please provide a pictorial diagram of the patient flow.

Author Response

Dear Sir or Madam:

Thank you for your kind and helpful comments. We have read through comments carefully and have made corrections. The responses to the reviewer’s comments and corrections are presented as following:

Point 1: In Introduction, the rationale for why cardioembolic stroke patients were selected for the study is insufficient.

Response 1: We thank the reviewer for the very helpful and constructive comment. Indeed, as we described in Line 39-48, resulting in more death and disability than other stroke subtypes and still lacking sufficient usage of optimized therapy (i.e. anticoagulant), identifying patients with potentially poor functional outcomes at an early stage would facilitate the selection of more intensive therapeutic interventions and ultimately benefit the prognosis of cardioembolic stroke patients in China. Secondly, as the progress of imaging techniques, SWI has now been gradually and widely used in our country, especially in west side of China, an area with relatively poor economic level and social development level. Secondly, as described in Line 49-61, DMVs in SWI has now been proved to be related in stroke. Furthermore, considering the rapid changes of pathophysiological process in acute cardioembolic stroke and a lack of research specifically focusing on such group, we proposed an observational study through retrospective data. Regarding this suggestion, we have added some new contents in introduction section, please see Line 65-66, 68-74 in revised version.

Point 2: Excluding patients who meet the exclusion criteria may bias the results of the analysis. How many patients meet each exclusion criterion? To clarify this, please provide a pictorial diagram of the patient flow.

Response 2: We are sorry for not being cautious and rigorous enough in the early version. In the revised version, we have added a flow chart of patient enrolment, please see Figure 2, page 4 in the revised version.

Thanks again for your kind and helpful comments.

Best Regards.

Reviewer 2 Report

Interesting work on the role of deep medullary veins (DMV) in the prognosis of patients after stroke. The authors assessed 170 patients with cardiovascular stroke who did not receive reperfusion treatment.

In the discussion, the authors discussed the effect of changes in the ischemic tissue and its consequences, including the contralateral hemisphere.

The authors hypothesized that severe contralateral DMVs may be associated with a relatively stronger collateral flow and a compensatory elevation of OEF to maintain the equilibrium of oxygen metabolism in the opposite hemisphere due to occlusion by a cardiac embolus.

This is interesting, but it should be noted that severe DMV (sDMV) occurred in the group:

1 / with a larger area of ​​cerebral infarction (in this group 13 [62%] patients ver 44 [29%] in the second group without sDMV),

2 / initially with a worse clinical condition - in the group with sDMV on mean NIHSS 15 versus mean NIHSS 8.

The results after 3 months were better were:

1 / in younger patients with lower NIHSS at baseline (5 ver 12),

2 / in patients with a smaller area of ​​damage (a large area in the 1st group is 26%, ver sDMV 73%),

3 / in the group where hemorrhagic transformation occurred less frequently, 35 ver 65%).

This would suggest that severe DMV is not a prognostic factor (i.e., venous blood flow disturbance and the size of the lesion / lesion area, and the degree of DMV is only a response proportional to the amount of brain damage.

This is an interesting problem, especially in the current era of intensive development of interventional stroke treatment - the question may arise whether it is worth giving small doses of thrombolytic, anticoagulants ?? to reduce the likelihood of venous dysfunction as a response to ischemia?

In my opinion, it is difficult to compare the results in patients in groups where the difference in the initial degree of neurological damage is so large, the spread of the size of the brain damage is so large, the frequency of hemorrhagic transformation is different, which has a huge impact on the long-term results.

My comments: In the group with a worse prognosis, 18 patients had sDMV - how many had a large area of ​​ischemia in the brain? Can you try to assess the influence of DMV on the frequency of haemorrhage transformation? It proposes to rebuild the discussion and supplement the results and clinical indicators.

Author Response

Dear Sir or Madam:

Thank you for your kind and helpful comments. We have read through comments carefully and have made corrections. The responses to the reviewer’s comments and corrections are presented as following:

We are deeply appreciated for the reviewer’s very helpful and constructive comments. Firstly, since the baseline difference between two severe DMVs groups are significant, especially for NIHSS, HT, and large infarction, all of which have impact on clinical outcomes, it is indeed inappropriate the define DMVs as “prognostic factor”. We deleted the descriptions using “prognostic” in the revised version and described as “associations”, please see Line 271-273, 342-346. Secondly, it has been well studied previously that prominent DMVs groups had higher admission NIHSS score and were prone to develop HT and large infarction size (AJNR, 2016. 37(3): p. 423-9; AJNR, 2014. 35(11): p. 2061-7; Sci Rep, 2021. 11(1): p. 5641), thus it is highly plausible that the degree of DMV is only a response proportional to the amount of brain damage. However, resulting from low drainage and enlargement of veins due to increased oxygen extraction fractions (OEF) in the hypoperfusion tissue and/or cytotoxic edema within the infarct area, and severe blood-brain barrier dysfunction (Cerebrovasc Dis, 2008. 26(4): p. 367-75; AJNR, 2011. 32(9): p. 1697-702; Eur J Radiol, 2012. 81(6): p. 1282-7), reflecting an increased level of deoxyhemoglobin, decrease of cerebral blood flow, poor establishment of collateral circulation (Stroke, 2002. 33(4): p. 967-71; Clin Neurol Neurosurg, 2009. 111(6): p. 483-95), severe DMVs may share common pathophysiological mechanisms with HT and large infarction, along with greater stroke severity and subsequently worse clinical outcomes, please see the added results of stratified analysis in Line235-241 and Table 4, page 8. Besides, subgroup analysis stratified by these potential confounders showed no interactions of these modifiers in the associations between DMVs and outcomes. Last but not the least, multivariable analysis after adjusting all confounders showed an independent association between DMVs and clinical outcomes (see Table 3, page 8). We have also amended the discussion part, please see Line 285-288, 298-308, page 9, and Line 309-316, page 10 in the revised version.

In addition, as the reviwer said, "in the current era of intensive development of interventional stroke treatment", therapeutic options need to be enriched for better clinical outcomes in the patients. Notably, effective treatments, such as thrombolysis, thrombectomy and anticoagulants, are accepted as standards of care across the world, including China in recent years, but bleeding-risk concerns still hamper widespread adoption of these care in Chinese people (The Lancet Neurology, 2019. 18(4): p. 394-405; Circ Cardiovasc Qual Outcomes, 2019. 12(12): p. e005610; Int J Cardiol, 2021. 322: p. 258-264). In the present study, there was a higher likelihood to have better clinical outcome for patients receiving anti-coagulation after admission (unadjusted OR 0.295, 95%CI 0.146 – 0.594, p=0.001), although there was no statistical significance in the multivariable anal-ysis (adjusted OR 0.393, 95% CI 0.154 – 1.006, p=0.051). It is proposed the use of more intensive therapy (such as thrombolysis, anticoagulants, etc.) in such patients after weighing benefits and risks, and it is needed for more clinical studies on venous dysfunction and anticoagulation in patients with acute ischemic stroke in the future. We have also added some contents about this point in the discussion part, please see Line 324-335, page 10 in the revised version.

Thank you and best regards.

Yours sincerely.

Reviewer 3 Report

Thank you for interesting study and clearly written manuscript. Your findings of severe deep medullary veins on SWI-MRI as predictor of poor functional outcome at 3 months after cardioembolic stroke seem convincing enough.

My concerns are about inter-rater reliability of the assessment of DMVs grade (eg, differentiating between DMV grade 2 and grade 3), based on visual inspection. You mentioned about the methods of visual assessment by two trained readers and the interrater kappa coefficient 0.88 for DMvs grade.  Nevertheless, could you please describe in more detail the rate of disagreement between evaluators and the most problematic situations in the practical evaluation of SWI imaging studies?

A few typos should be corrected (eg, lines 261-262: "We hypothesis" -> "We assume" or "We hypothesize".

Author Response

Dear Sir or Madam:

Thank you for your kind and helpful comments. We have read through comments carefully and have made corrections. The responses to the reviewer’s comments and corrections are presented as following:

Point 1: My concerns are about inter-rater reliability of the assessment of DMVs grade (eg, differentiating between DMV grade 2 and grade 3), based on visual inspection. You mentioned about the methods of visual assessment by two trained readers and the interrater kappa coefficient 0.88 for DMvs grade.  Nevertheless, could you please describe in more detail the rate of disagreement between evaluators and the most problematic situations in the practical evaluation of SWI imaging studies?

Response 1: We thank the reviewer for the very helpful and constructive comment. More details about interpretations of imaging indicators are added in Line 141-145, page 3 in the revised version. The most problematic situations in the practical evaluation of SWI imaging studies were the unsatisfactory image quality and inter-researcher interpretation consistency. To make accurate assessment of DMVs grade, we strictly excluded patients with obvious head movements, artifacts, midline shift, intracranial mass effect or other conditions that will result in poor image quality and affect the interpretation of DMVs in the final analysis (please see Figure 2, page 4 in the revised version). A disagreement was solved through discussion or advice from an expert who has vast research and clinical experience in the field of neuroimaging and published articles in top academic journals (Neurology, 2015, 85:2045–2052; Stroke, 2020, 51(9):2801-2809, etc.). Besides, all patients’ MR images were acquired through a same MR machine (Trio Siemens, Berlin, Germany) with a set of same scanning parameters to minimize heterogeneity due to objective causes.

Point 2: A few typos should be corrected (eg, lines 261-262: "We hypothesis" -> "We assume" or "We hypothesize".

Response 2: Thanks for the reviewer’s constructive advice. It’s our mistake for not being cautious enough. We amended and corrected these wrong typos in the revised version.

Thank you and best regards.

Yours sincerely.